# OsmiRNA5488 Regulates the Development of Embryo Sacs and Targets *OsARF25* in Rice (*Oryza sativa* L.)

**DOI:** 10.3390/ijms242216240

**Published:** 2023-11-13

**Authors:** Shengyuan Guo, Chuanjiang Zheng, Yan Wang, Yangwen Xu, Jinwen Wu, Lan Wang, Xiangdong Liu, Zhixiong Chen

**Affiliations:** 1Department of Plant Breeding, College of Agriculture, South China Agricultural University, Guangzhou 510642, China; 20213137020@stu.scau.edu.cn (S.G.); 2469214490@stu.scau.edu.cn (C.Z.); 20213137065@stu.scau.edu.cn (Y.W.); 20223137111@stu.scau.edu.cn (Y.X.); jwwu@scau.edu.cn (J.W.); wanglan@scau.edu.cn (L.W.); 2State Key Laboratory for Conservation and Utilization of Subtropical Agro-Bioresources, South China Agricultural University, Guangzhou 510642, China; 3Guangdong Provincial Key Laboratory of Plant Molecular Breeding, South China Agricultural University, Guangzhou 510642, China; 4Guangdong Laboratory for Lingnan Modern Agriculture, South China Agricultural University, Guangzhou 510642, China

**Keywords:** rice, miRNA, seed-setting rate, embryo sac, IAA

## Abstract

Small RNAs are a class of non-coding RNAs that typically range from 20 to 24 nucleotides in length. Among them, microRNAs (miRNAs) are particularly important regulators for plant development. The biological function of the conserved miRNAs has been studied extensively in plants, while that of the species-specific miRNAs has been studied in-depth. In this study, the regulatory role of a rice-specific OsmiRNA5488 (OsmiR5488) was characterized with the miR5488-overexpressed line (*miR5488-OE*) and miR5488-silenced line (*STTM-5488*). The seed-setting rate was notably reduced in *miR5488-OE* lines, but not in *STTM-5488* lines. Cytological observation demonstrated the different types of abnormal mature embryo sacs, including the degeneration of embryo sacs and other variant types, in *miR5488-OE* lines. The percentage of the abnormal mature embryo sacs accounted for the reduced value of the seed-setting rate. Furthermore, *OsARF25* was identified as a target of *OsmiR5488* via RNA ligase-mediated 3′-amplifification of cDNA ends, dual luciferase assays, and transient expression assays. The primary root length was decreased with the increases in auxin concentrations in *miR5488-OE* lines compared to wild-type rice. Summarily, our results suggested that OsmiR5488 regulates the seed-setting rate and down-regulates the targeted gene *OsARF25*.

## 1. Introduction

The seed-setting rate of rice is affected by a variety of genetic and environmental factors, including defects in embryo sac development, malformation of floral organs, defects in pollen grain formation, insufficient anther dehiscence, gametophytic incompatibility, and temperature adversity [1]. So far, considerable progress has been made in the gene regulation of pistil development in rice. Mutations in the rice B gene *OsMADS6* or the C gene *DL* resulted in the homeotic transformation of stamens or carpel, respectively [2,3]. A mutation in the class D gene *OsMADS13* led to the transformation of somatic ovules into carpel structures [4]. Furthermore, the floral meristems of *fon1* and *fon2/fon4* mutants were enlarged and caused an increase in the number of floral organs in all four whorls, including carpels [5,6]. A majority of ovaries produced double ovules in *OsIG1-RNAi* transgenic lines [7]. Mutation in *LOG* caused abnormal florets without a pistal [8]. Additionally, most ovaries of *ospid* mutants were developed with the absence of ovules [9]. The female gametophyte is formed after megesporogenesis and megagametogenesis. In *Arabidopsis*, numerous genes are involved in the two distinct processes [9]. However, a few genes have been identified to regulate female gametogenesis in rice. Abnormality in functional megaspore mitosis was observed in *OsAPC6* mutants [10]. In *OsDEES1-RNAi* plants, the megaspores developed normally, but female gametogenesis was disrupted, resulting in the formation of degenerated embryo sacs [11]. *esd1* mutant produced the degeneration of some egg cells and caused a decrease in seed-setting rate [12]. In *OsROS1-Cas9* mutants, embryo sac degeneration was characterized by the absence of cavities or nuclei [13]. The functional analysis of these protein-coding genes contributes to the understanding of the molecular regulation mechanism involved in the pistil development of rice.

MicroRNA (miRNA), a small non-coding RNA molecule, is precisely processed from hairpin precursors and represses the functions of targeted genes at post-transcriptional or translational level [14,15,16]. MiRNAs were proven as important regulators of various cellular processes in plants [17]. AtmiR160-AtARF17 module orchestrated auxin signaling to contribute to the specification of the female germline [18]. Null *miR167a* mutations caused defective anther dehiscence and ovule development [19]. During the young inflorescence development of rice, a total of 111 miRNAs were characterized as differentially expressed miRNAs (DEMs), including OsmiR5488 [20]. A total of 56, 65, and 11 DEMs were identified in the ovule during the meiosis process, megaspore mitosis, and mature embryo sac stage, respectively [14]. A total of 368 DEMs, including OsmiR5488, were found during embryo sac development in autotetraploid rice, respectively [21]. That transcriptome analysis highlighted the important regulatory roles of miRNAs in rice reproductive development. Rice OsmiR5488 is a species-specific miRNA. Its targets are still uncharacterized for now. In this study, experimental results showed that the overexpression of OsmiR5488 led to a decrease in seed-setting rate. These findings suggested that OsmiR5488 may play a significant role in regulating the seed-setting rate.

Auxin, a vital phytohormone, plays a critical role in the growth and development of plants. *Auxin response factors* (*ARFs*) exert their roles by regulating the auxin-mediated pathway. In *Arabidopsis*, *ARF5*/*MP, NONPHOTOTROPIC HYPOCOTYL4* (*NPH4*)*/ARF7, BODENLOS/IAA12,* and *AUXIN RESISTANT3 (AXR3)/IAA17* are involved in regulating primary root formation [22]. Additionally, the *ARF5/MP* gene governs the initiation of embryonic roots by controlling a mobile transcription factor called *TARGET* of *MP 7* (*TMO7*) [23]. During the development of the female gametophyte in *Arabidopsis*, the simultaneous down-regulation of *AtARF2*-*AtARF4* brought about abnormal embryo sacs, exhibiting identity defects, such as the formation of two cells with an egg cell-like morphology, concomitant with the loss of synergid marker expression and seed abortion [14]. In the rice genome, a total of 25 *ARF* genes are annotated [24]. Knocking out *OsARF12* resulted in a decrease in the length of the primary root, and the root elongation zones of *osarf12* and *osarf12/25* were significantly shorter and presented lower auxin concentrations compared to that of wild-type rice [25]. *OsARF23-OsARF24* promotes cell growth and morphogenesis by regulating the expression of RICE morphological determinants (RMDs) [26]. *OsARF4* regulates leaf inclination via auxin and brassinosteroid pathways [27] and interacts with *OsGSK5/OsSK41* to negatively modulate rice grain size and weight [28]. Transcriptome analysis identified that *ARF* genes might be involved in rice fruit development [29]. However, the functions of *ARFs* involved in reproductive growth and development have not been studied extensively in rice. In this study, we functionally characterized a rice-specific OsmiR5488 in regulating the seed-setting rate and demonstrated that OsmiR5488 targeted *OsARF25.*

## 2. Results

### 2.1. OsmiR5488 Is Ubiquitously Expressed in Organs

The species-specific miRNAs usually exhibit low but organ or tissue-specific expression, and a few of them might have obtained specialized functions during evolution [30]. The expression pattern of OsmiR5488 was investigated using stem-loop qRT-PCR. It was suggested that OsmiR5488 transcripts were highly expressed in roots and pollen, but comparatively lower in other organs tested from the vegetative to reproductive stages (Figure 1), implying that OsmiR5488 might have regulatory roles in reproductive development or root growth.

### 2.2. The Phenotypic and Expression Changes in OsmiR5488 Transgenic Lines

To gain insight into the biological function of OsmiR5488, the miR5488-overexpressed line (*miR5488-OE*) was generated by inserting the pre-miR5488 sequence fragment between the *Ubiquitin* promoter and *Nos* terminator of the *POX* construct. Meanwhile, the miR5488-silenced line (*STTM-5488*) was obtained by overexpressing the STTM fragment containing two copies of imperfect miR5488 binding sites (24 nucleotides) linked to both flanks of a 48-nucleotide RNA spacer under the control *Ubiquitin* promoter (Appendix A). During the vegetative growth stage, both types of transgenic lines displayed the normal phenotype as wild-type (WT) Nip (Figure 2A). After grain maturity, the seed-setting rate was significantly lower in *miR5488-OE*, but not changed in *STTM-5488*, compared to that of WT (Figure 2B). There were no significant differences in grain length, grain width, and grain thickness among either of the two types of transgenic lines and WT (Appendix A). QRT-PCR analysis revealed that the expression level of OsmiR5488 was largely enhanced in *miR5488-OE* and decreased in STTM-miR5488 lines (Figure 2C). It was suggested that OsmiR5488 might play a vital role in reproductive development and growth.

### 2.3. Abnormalities in Mature Embryo Sacs in OsmiR5488 Transgenic Lines

To find the reason underlying the low seed-setting rate of *miR5488-OE* lines, we firstly applied I_2_-KI solution to check the fertility of pollens. There was no obvious difference in the fertility of pollen grains among WT, *miR5488-OE*, and *STTM-5488* lines (Figure 3A–E,P). Secondly, pollen viability was examined using the Alexander stain, which distinguishes viable pollen grains from dead ones. No obvious difference in the viability of pollen grains was detected among the tested lines (Figure 3F–J,Q). Thirdly, whole-mount eosin B-staining confocal laser scanning microscopy (WE-CLSM) observation showed that the normal mature embryo sac was characterized by one egg, two synergids, two polar nuclei, and three antipodal cells in both wild-type and *miR5488-OE* lines (Figure 3K). Moreover, the different types of abnormalities in ovaries were found in the *miR5488-OE* lines, including the degeneration of the embryo sac, abnormal number of polar nuclei, embryo sac with degenerated egg apparatus, and a small embryo sac (Figure 3L–O). In contrast, *STTM-5488* did not show an abnormal embryo sac (Figure 3K). The percentage of abnormal embryo sacs amounted to 48% in *miR5488-OE-1* and 53% in *miR5488-OE-2* lines, while that in WT rice only reached 4%. The degeneration of the embryo sac was the major abnormal type in *miR5488-OE* lines. It was indicated that OsmiR5488 affects the seed-setting rate in rice by regulating the development of embryo sacs.

### 2.4. OsARF25 Was Down-Regulated by OsmiR5488

It is known that plant miRNA regulates development and growth by down-regulating the targeted genes. It is predicted that *OsARF25* (*LOC_Os12g41950*) is one of the genes targeted by OsmiR5488 through psRNATarget (https://www.zhaolab.org/psRNATarget/, accessed on 17 November 2021) and ENCORI (https://rnasysu.com/encori/ accessed on 20 November 2021) tools. To elucidate whether *OsARF25* is the authentic target gene of OsmiR5488, we examined the *OsARF25* transcript abundance in the root tissue and embryo sacs of *miR5488-OE* and *STTM-5488* lines, respectively, using qRT-PCR. It was indicated that the expression level of *OsARF25* was down-regulated in the root of *miR5488-OE* lines and up-regulated in STTM5488 lines, respectively, compared to that of the wild type (Figure 4A). Similar changes in *OsARF25* expression were found in the embryo sacs of both transgenic lines (Figure 4B). It was suggested that the expression level of *OsARF25* was negatively regulated by OsmiR5488.

### 2.5. OsARF25 Is Targeted by OsmiR5488

To further verify the target relationship between OsmiR5488 and *OsARF25*, 3′ rapid amplification of cDNA ends (3′-RACE) was carried out to map the OsmiR5488-directed guide sites in the *OsARF25* transcript sequence. It was shown that the last exon at the 3′-UTR region of *OsARF25* mRNA was precisely cleaved, and the cleavage site was proven to be the base pairs between the eighth and ninth nucleotide of the targeted sequence (Figure 5A). Secondly, the LUC reporter system was further used to verify the cleavage site of *OsARF25*. The target site of *OsARF25* was mutated (*OsARF25mts*) by inserting six-base nucleotides between the cleavage sites (Appendix A). The fragment containing the normal 21 bp target sequence (*OsARF25ts*) and its mutant version (*OsARF25mts*) were inserted into a reporter vector downstream of firefly LUC (FLUC), respectively (Figure 5B). Renilla LUC (RLUC) was used as an internal control. The OsmiR5488 construct was applied as an effector vector. The quantification of LUC activity showed that FLUC/RLUC values were significantly reduced by the co-transformation of *35S::OsmiR5488* and *OsARF25ts-FLUC*, but were not altered by the combination of *35S::OsmiR5488* and *OsARF25mts-FLUC* (Figure 5C). Taken together, these findings suggest that OsmiR5488 could target *OsARF25* mRNA in vivo.

We designed a GFP-based reporter assay to verify the cleavage site, too. Given that the target bases of OsmiR5488 were located in the 3′-UTR of *OsARF25*, we designed CaMV 35S-driven constructs, in which GFP was fused to a truncated *OsARF25* with a normal OsmiR5488 target site (*35S::OsARF25ts*) or with a mutated OsmiR5488 target site (*35S::OsARF25mts*), respectively (Appendix A). The constructs were transiently co-expressed with *35S::miR5488* in *Nicotiana benthamiana* leaves by *Agrobacterium tumefaciens*-mediated transformation. GFP expression was determined by fluorescence intensity. When expressed alone, *35S::GFP*, *35S::OsARF25ts*, and *35S::OsARF25mts* emitted highly abundant fluorescence intensity, respectively (Figure 5D,E). However, when co-expressed with *35S::miR5488*, the fluorescence intensity of *35S::OsARF25ts* was significantly reduced, especially at high concentrations of the infiltrated *Agrobacterium*, while that of *35S::OsARF25mts* was not changed (Figure 5D,E). It is therefore evident that OsmiR5488 suppresses *OsARF25* expression by targeting the *OsARF25* 3′-UTR. In summary, these results strongly demonstrated that *OsARF25* is targeted by OsmiR5488 in vivo.

### 2.6. miR5488-OE Exhibited Less Sensitivity to IAA Treatment

Previous studies have proven that auxin homeostasis is a key factor in regulating pistil formation in plants [31]. We determined *OsARF25* as the downstream target of OsmiR5488. Thus, *miR5488-OE* lines were selected to analyze the response of OsmiR5488 to IAA phytohormone. The germinated seeds of *miR5488-OE* and WT were treated with IAA treatments of different concentrations. It was indicated that the length of the primary root was increased at 0.1 μM IAA, and decreased slowly at 1 μM IAA, and declined sharply at 10 μM IAA in the WT line (Figure 6A,C). On the contrary, the primary root length decreased gradually with the increase in IAA concentration in *miR5488-OE* lines (Figure 6B,C). Especially, the primary root length of *miR5488-OE* lines was significantly shorter than that of WT lines under a low concentration of IAA, but longer under the highest concentration of IAA (Figure 6C). It was suggested that *miR5488-OE* lines were less sensitive to IAA compared to wild-type Nip.

## 3. Discussion

Rice miRNAs play a direct or indirect role in influencing the four important yield components, including seed-setting rate, tiller number, grain number per panicle, and thousand-grain weight [32]. Many of the target genes by miRNAs are transcription factors, and some miRNAs exhibit pleiotropic effects [33]. For instance, the OsmiR156-*OsSPL14* module negatively regulates tiller formation [34], while the OsmiR156-*OsSPL13* module positively regulates grain size and branch number [35]. Additionally, OsmiR397, OsmiR396, and OsmiR1432 have been found to positively regulate yield-related traits [36,37,38]. Furthermore, OsmiR2118, OsmiR1848, and OsmiR528 indirectly influence the seed-setting rate by negatively regulating pollen fertility [39,40,41]. Meanwhile, there is limited experimental evidence on the functions of rice species-specific miRNAs. Rice-specific miR1848 exerts an impact on pollen fertility [42]. Rice-specific OsmiR5506 plays an essential role in the regulation of floral organ number, spikelet determinacy, and female gametophyte development in rice [20]. Rice OsmiR5488 is a rice-specific miRNA, too. The high expression abundance of OsmiR5488 was investigated in vegetative or reproductive organs (Figure 1). The expression pattern of OsmiR5488 was contrary to the low but organ or tissue-specific expression of the species-specific miRNAs [30]. It is implied that OsmiR5488 might obtain some functions in regulating the normal growth of plants during evolution. *miR5488-OE* lines were generated and showed a significant decrease in seed-setting rate (Figure 2B). Moreover, the abnormalities of mature embryo sacs were uncovered by WE-CLSM in *miR5488-OE* lines (Figure 3L–O), and its percentage almost amounted to the decreased value of seed-setting rate (Figure 2B and Figure 3R), indicating the cause for the low seed-setting rate in *miR5488-OE*. The defects of mature embryo sacs were also investigated in transgenic lines, overexpressing another rice-specific miR5506 [20], but some types of abnormalities were different from those in *miR5488-OE*. It was suggested that more than one rice-specific miRNAs, including OsmiR5488, were involved in the development of embryo sacs, and that miRNAs might act in overlapping or redundant patterns.

The development of female gametophytes is a complicated progress, which is regulated by many phytohormone-related genes and signal pathways. During the development of rice female gametophytes, the expression levels of auxin signaling-related genes (*OsARF6, OsARF18, OsARF22, OsARF25, OsIAA24*, and *OsIAA30*) significantly increased from meiotic diplotene to maturity [1]. Additionally, during the transition from meiosis to mitosis, the expression levels of auxin transport-related genes (*OsPIN1a, OsPIN1b, OsPIN1c* and *OsPIN1d*) were significantly down-regulated, while auxin synthesis-related genes (*OsYUC3* and *OsYUC9*) showed high expression during meiosis [40]. Furthermore, more than 40 auxin signaling-related genes exhibited ovule-specific expression in the early stage of female gametophyte development [43]. The concentrations of auxin in the pistil and stamen of the rice ovule-less mutant *ospid* were found to be reduced, and there was also a decrease in the expression level of most *ARF* genes. The regulators involved in governing the change in auxin concentrations or signaling should be elucidated. In *Arabidopsis*, miR167 controls patterns of *ARF6* and *ARF8* expression and regulates both female and male reproduction [44]. The AtmiR160-targeted gene *ARF17* is required for promoting the specification of megaspore mother cells by genetically interacting with the *SPOROCYTELESS/NOZZLE* gene [19]. OsmiR5506 down-regulates the expression level of *REM16* transcriptional factor and hormone-related genes [20]. Using three different methods, *OsARF25* was confirmed as the target of OsmiR5488 (Figure 4A and Figure 5A). *osarf25* was sensitive to 1 μM 2,4-dichlorophenoxyacetic acid treatment, but *osarf12/25* was less sensitive to it [25]. Thus, the auxin response of OsmiR5488 was analyzed preliminarily with different concentrations of IAA. It was revealed that *miR5488-OE* lines were less sensitive to IAA, in contrast to wild-type Nip (Figure 6), indicating that OsmiR5488 might be involved in the auxin signal pathway. It was necessary to further investigate the response of OsmiR5488 to different types of auxin and to compare the development of embryo sacs between *miR5488-OE* and *osarf25* mutants. In conclusion, rice-specific OsmiR5488 affects the seed-setting rate by regulating the development of embryo sacs and it down-regulates the targeted gene *OsARF25,* which is involved in auxin signaling.

## 4. Materials and Methods

### 4.1. Plant Materials and Growth Conditions

To overexpress miR5488, the fragment of pri-miR5488 was amplified and cloned into *POX* vector, generating the *miR5488-OE* lines. To generate the knock-down of the miR5488 line (*STTM-5488*), a 48-bp oligonucleotide was utilized as a linker to construct STTM miRNA, and the full-sequence STTM miRNA was amplified and cloned into a *POX* vector according to the methods in [40]. The constructs were introduced into the genome of wild-type (WT) variety Nipponbar (NIP) with *Agrobacterium*-mediated transformation [45]. The sequences and restriction sites of all primers and DNA fragments used for vector construction are listed in Appendix A. 

### 4.2. Treatment and Analysis of Root Growth

Germinated rice seeds were placed on a stainless steel sieve that was placed in a plastic box of 1 L volume and incubated at 28 °C. The seeds were grown in Kimura B complete nutrient solution supplemented with 0.1, 1, or 10 μM IAA, respectively. IAA was dissolved in ethanol. The controls were conducted with treatments containing equivalent volumes of ethanol. After 4-day treatment, the roots were scanned and their length was measured from digitized images using Image J software (version 6.0).

### 4.3. RNA Extraction and RT–qPCR Analysis

For qRT-PCR analysis, different tissues at different development stages were harvested from WT and transgenic lines, which were planted in the experimental farms in South China Agriculture University, Guangzhou City, Guangdong Province, China. The harvested tissues were immediately snap-frozen in liquid nitrogen and stored at −80 °C. RNA extraction and reverse transcription were carried out using the methods described by Zhao et al. [46]. The qRT-PCR reactions were conducted using UltraSYBR Mixture (YEASEN, Shanghai, China) and Roche Lightcycler 480 (Roche, Mannheim, Germany). Primers used for qRT-PCR analysis are listed in Appendix A.

### 4.4. Rapid Amplification of cDNA 3′ Ends (3′-RACE) Assays

3′-RLM-RACE was performed with the SMART RACE kit (Clontech Dalian, China) to identify the cleavage site of miR5488 in *OsARF25*, according to a modified user manual. Total RNA was extracted from miR5488 overexpression transgenic plants. The gDNA was cleaned with the Evo M-MLV RT Kit with gDNA Clean for qPCR (Accurate Biology, Changsha, China). The 3′ RACE adapter was ligated to the degraded mRNAs with T4 RNA ligase. Reverse transcription was at 42 °C for 1 h with 5 units M-MLV reverse transcriptase and 25 pmol antisense RNA specific primer (3′ RACE RT). The cDNA was amplified with an adaptor-specific primer (3′ RACE outer primer) and a specific primer (3′ RACE GSP1). A second amplification was performed with 3′ RACE inner primer and 3′ RACE GSP2 using the first PCR products as a template. Purified PCR products were cloned into a pGEM-T vector. Bacterial colonies were checked for appropriate inserts using PCR and confirmed through sequencing.

### 4.5. Dual LUC Transient Expression System

The wild-type 3′-UTR sequences of *OsARF25* (*OsARF25ts*) and mutant ones (*OsARF25mts*) were inserted into the pGreen II 0800 dual-luciferase reporter vector, respectively. All plasmids were transferred into *Agrobacterium tumefaciens* GV3101, and the *A. tumefaciens* strains were cultured and suspended with infiltration solution. The prepared effector and reporter mixtures were injected into *Nicotiana benthamiana* leaves. After three days of injection, the infiltrated *N. benthamiana* leaves were applied to quantify the activities of FLUC and RLUC, using SpectraMax iD3 Multi-Mode Microplate Readers (Molecular Devices) and the Dual-Glo LUC Assay System (Promega, Madison, WI, USA). At least five replicates were performed for the quantification of FLUC/RLUC values.

### 4.6. Transient Expression Assays

The GFP fused to truncated *OsARF25* with the normal OsmiR5488 target site (*35S::OsARF25ts*) or with the mutant OsmiR5488 target site (*35S::OsARF25mts*) was inserted into the CaMV 35S-driven vector. Relevant vectors were transformed into GV3101. Cultured cells were centrifuged at 4000× *g* for 5 min at room temperature. The harvested bacterial cells were resuspended and diluted to the appropriate OD600 with infiltration buffer (10 mm MES, 10 mm MgCl_2_, and 0.2 mm acetosyringone, pH 5.6). The resuspended bacteria were infiltrated into the abaxial side of 5–6-week-old *N*. *benthamiana* leaves. The samples were observed with a Leica SP2 laser scanning confocal microscope (Leica Microsystems, Heidelberg, Germany). 

### 4.7. Pollen Analysis

To evaluate mature pollen fertility, WT and mutant anthers were collected and stained with 1% (*w*/*v*) iodine–potassium iodide solution (I_2_-KI), and the accumulation of starch in pollen grains was observed using a Leica DM2000 microscope (Leica, Wetzlar, Germany). Anthers were stained in Alexander solution to stain pollen grains and observed with a Leica DM2000 microscope [47].

### 4.8. Eosin B Staining and Embryo Sac Scanning

Eosin B staining and embryo sac scanning were based on the method of Zeng et al. [48]. The ovaries were dissected in 70% ethanol under a binocular dissecting microscope, and hydrated sequentially in 50% ethanol, 30% ethanol, and distilled water. Then, the ovaries were pretreated in 2% aluminum potassium sulfate for 20 min to allow the dye to enter the embryo sac more readily. Then, the ovaries were stained with 10 mg/L eosin B (C_20_H_6_N_2_O_9_Br_2_Na_2_, FW 624.1, a tissue stain for cell granules and nucleoli) solution (dissolved in 4% sucrose) for 10–12 h at room temperature. The samples were post-treated in 2% aluminum potassium sulfate for 20 min to remove some dye from the ovary walls. The samples were rinsed with distilled water three times, and dehydrated with a series of ethanol solutions (30%, 50%, 70%, 90%, and 100%). Subsequently, the dehydrated samples were transferred into a mixture of absolute ethanol and methyl salicylate (1:1) for 1 h and then cleared in pure methyl salicylate solution for at least 1 h. The samples were scanned under a Leica SP2 laser scanning confocal microscope (Leica Microsystems, Heidelberg, Germany). The excitation wavelength was 543 nm, and emitted light was detected between 550 and 630 nm. The images of different focal planes of a sample were recorded [48].

### 4.9. Statistical Analyses

SPSS 22.0 software (SPSS, USA) was used to perform statistical analyses, and the mean values were presented with a standard deviation of four biological replicates, and were compared via Student’s *t*-test at the probability level of *p* < 0.05 or 0.01. All graphs were created using the GraphPad software (version 8.0).

## Figures and Tables

**Figure 1 ijms-24-16240-f001:**
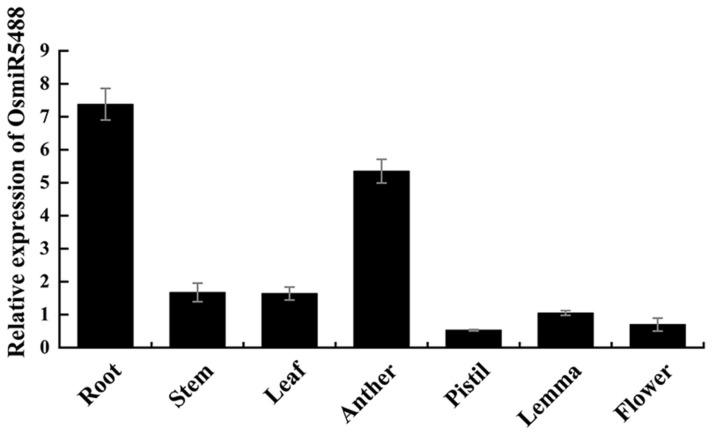
qRT-PCR analysis of relative expression levels of OsmiR5488 in different tissues from wild-type Nipponbare (NIP). Data are mean ± s.e.m. (*n* = 3 plants each with three technical replicates).

**Figure 2 ijms-24-16240-f002:**
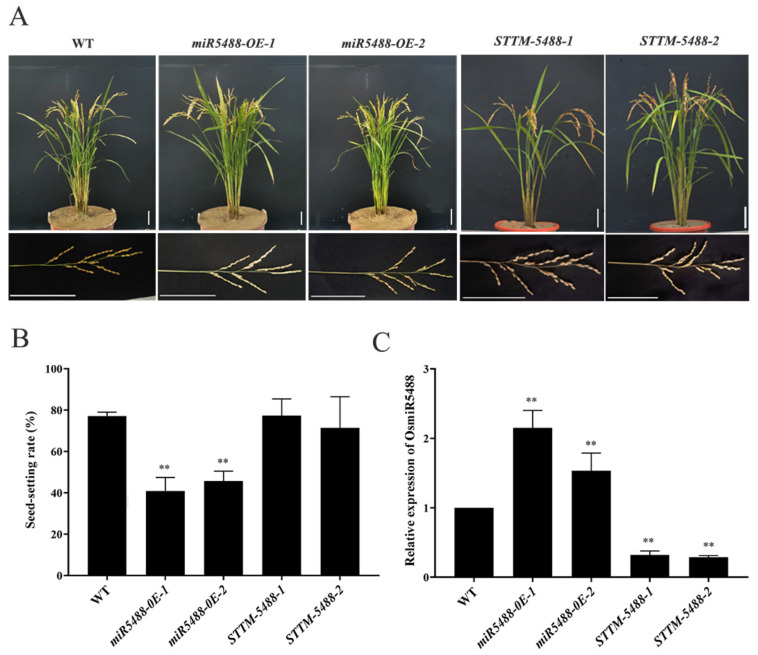
Phenotype, expression level of OsmiR5488, and seed-setting rate of the transgenic lines of *miR5488-OE* and *STTM-5488*. (**A**) Similar phenotypes among WT, *miR5488-OE*, and *STTM-5488* plants at the mature growth stage. (**B**) Seed-setting rate of WT, *miR5488-OE*, and *STTM-5488* lines. Data are means (±s.e.m.) (*n* = 20 plants, **, *p* < 0.01). (**C**) The change in expression level of OsmiR5488 in embryo sacs of WT, *miR5488-OE*, and *STTM-5488* lines.

**Figure 3 ijms-24-16240-f003:**
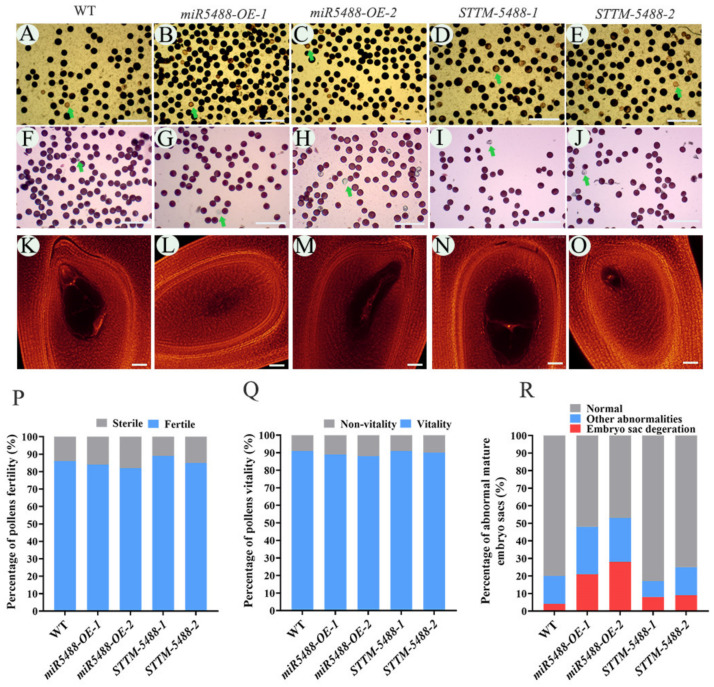
Analysis of the fertility of mature pollens and embryo sacs in miR5488 transgenic lines. (**A**–**E**) Pollen fertility; green arrow indicated the sterile pollen; bar = 10 μm. (**F**–**J**) Pollen vitality; green arrow indicated the non-vitalitious pollen; bar = 10 μm. (**K**) Normal embryo sac. (**L**) Degeneration of embryo. (**M**) Abnormal number of polar nuclei. (**N**) Abnormal embryo sac with degenerated egg apparatus. (**O**) Small embryo sac. Bar = 40 μm in (**K**–**O**). (**P**) Percentage of pollen fertility. (**Q**) Percentage of pollen vitality. (**R**) Percentage of normal or abnormal embryo sac.

**Figure 4 ijms-24-16240-f004:**
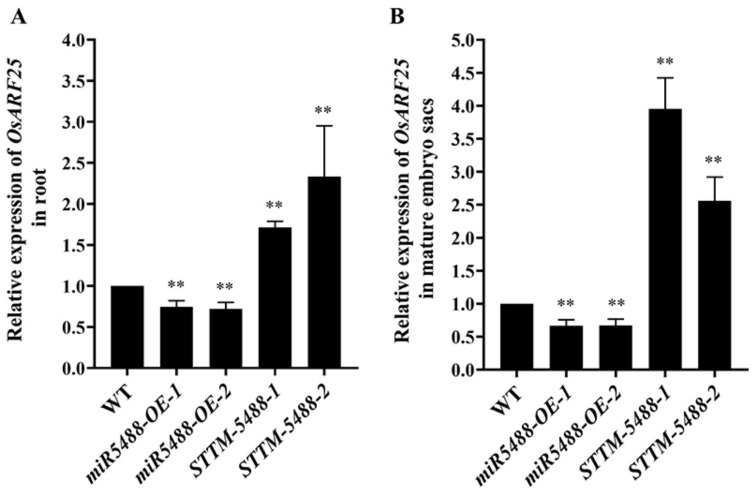
The down-regulation of *OsARF25* by OsmiR5488. (**A**) The change in expression levels of *OsARF25* in root. (**B**) The change in expression levels of *OsARF25* in mature embryo sac. Asterisks indicated significant differences by Student’s *t*-test, **, *p* < 0.01.

**Figure 5 ijms-24-16240-f005:**
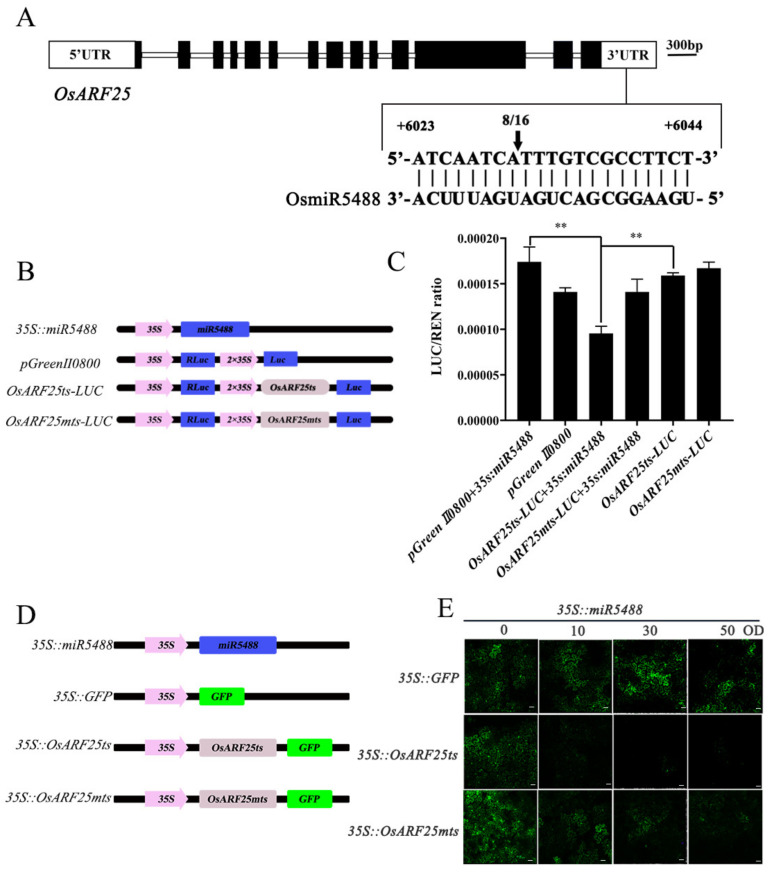
The identification of *OsARF2*5 as a targeted gene by OsmiR5488. (**A**) Gene structures of *OsARF25* and the cleavage site of *OsARF25* confirmed by sequencing 8 clones out of 16. (**B**) Schematic diagram of OsmiR5488 or *OsARF25* constructs for dual LUC assay. (**C**) Dual LUC assays indicated that *OsARF25* was targeted by OsmiR5488. The RLUC/FLUC values were obtained with at least five replicates. **, *p*  <  0.01 by Student’s *t* test. (**D**) Schematic diagram of OsmiR5488 or *OsARF25* constructs for transient expression assays. (**E**) The GFP transient expression assays showed that *OsARF25* was targeted by OsmiR5488. Bars: 40 μm.

**Figure 6 ijms-24-16240-f006:**
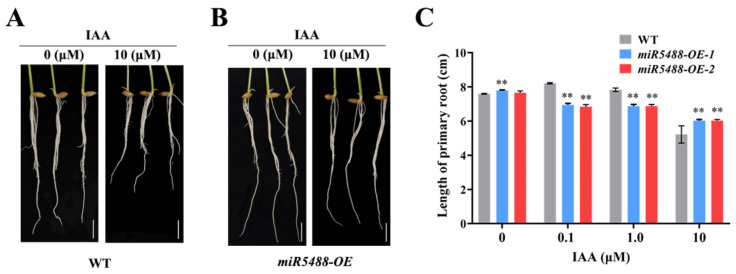
Comparison of the response of primary root length to IAA between *miR5488-OE* and wild-type Nip lines. (**A**) Primary root length of Nip after 4-day treatment with 10 μM IAA. Bar, 1 cm. (**B**) Primary root length of *miR5488-OE* after 4-day treatment with 10 μM IAA. Bar, 1 cm. (**C**) Statistical comparison of primary root length between WT and *miR5488-OE* after 4-day treatment with different concentrations of IAA. Values were shown as the mean ± SD of 20–30 seedlings, The experiment was repeated at least three times with similar results, **, *p* < 0.01.

## Data Availability

Data is contained within the article or
Appendix A.

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
