# Peer review of "OsmiRNA5488 Regulates the Development of Embryo Sacs and Targets *OsARF25* in Rice (*Oryza sativa* L.)"

_ijms, 2023, doi:10.3390/ijms242216240_

Round 1

Reviewer 1 Report (Previous Reviewer 2)

Comments and Suggestions for Authors

Authors have clearly improved their manuscript, but not all modifications were highlighted nor described in the cover letter, making very very difficult to locate modifications. After this detective effort, I am not considering the manuscript relevant for IJMS due to the following flaws after the deep re-thinking explained by the authors:

1) The response to issue #3 is not valid. Authors mention in Introduction that their miRNA is a DEM, but do not give any clue about why this miRNA is more important than the other 56, 65 and 11 DEMs. Their rationale in the cover letter is that this article reveals that it is important, but my question is what drives authors to study this miRNA and not the other DEMs.

2) Figure 2 is still a nightmare. IT has only 3 pannels, but the figure legend describes up to (an absent) panel D. IN fact, the figure has been changed from the initial manuscript, but the legend was maintained. It is not described what are the parameters resumed under the title "seed setting rate" in this figure. The relative expression is relative to what? I am still unable to understand what this figure es presenting (I can only guess the intention of authors). Moreover, possible ideas to understand it are on the cover letter but not in the manuscript. Hence, any future reader may have the same concerns than myself, making it unacceptable in the current form.

3) Concerning point 6 in the cover letter, I was unable to detect the changes without spending an unnecessary time looking for differences among PDFs. In fact, I detected many many unnoticed changes with respect to the previous manuscript regarding text, figures, headers, etc.

4) Concerning the sentence in lines 144-145 "The percentage of abnormal ovaries amounted to 48% in miR5488-OE-1 and 53% in miR5488-OE-2 lines, respectively", are these values statistically different? I do now think so because the difference in % is very small. Hence, maybe this miRNA is not affecting the ovary development.

5) Lines 157-160: authors must present numeric evidences that demonstrate that LOC_Os12g41950 is the "authentic target" of their miRNA. Moreover, use only one name for genes, since in the same sentence they indicate that the LOC gene is also called OsARF25, which is the name preferentially used in the manuscript.

6) In Fig. 5, it is not explained what is the "relative expression". This is absolutely required to know whether the expression of the interesting gene increases or decreases.

7) Concerning section 2.5, a miRNA cannot cleave any mRNA. Hence, this section lacks of scientific support. Perhaps authors would like to say that the miRNA guides/marks/selects the mRNA for cleavage. The precise cleavage site described is guided by the miRNA, but not cleaved by the miRNA.

8) The discussion is a mere review of what are miRNAs doing in their model with only a few words relating to miR5488. This is an important issue about the relevance of this work.

9) English review is still required. I detected many errors even if I am not a native speaker.

Comments on the Quality of English Language

There are many mistakes. Most of them can be corrected using an automatic grammar checker. Some sentences are understandable although they seem bizarre.

Author Response

Dear Reviewers,

Thanks very much for taking your time to review this manuscript. I really appreciate all your comments and suggestions! Please find my itemized responses in below and my revisions/corrections in the re-submitted files.

1) The response to issue #3 is not valid. Authors mention in Introduction that their miRNA is a DEM, but do not give any clue about why this miRNA is more important than the other 56, 65 and 11 DEMs. Their rationale in the cover letter is that this article reveals that it is important, but my question is what drives authors to study this miRNA and not the other DEMs.

Response: Thanks for your kind suggestions. Though miRNAs emerge as important regulators for plant growth and development. Many DEMs, including conserved and non-conserved miRNAs, were identified by transcriptome analysis. The conserved DEMs were often selected to study their function with good publication. While the non-conserved miRNAs were predicted to have low expression level or tissue (or organ) -specific expression pattern and are worth further study. miR5488, was identified as DEMs in inflorescence development by different researchers groups, as well as with some other miRNAs. It is difficult to compare the importance of miRNA to other DEMs, based on the data available. We only presumed that miR5488 might be vital regulator in rice. In our lab, the function of some non-conserved DEMs are being studied by overexpression transgenic lines, besides miR5488. It was found that some non-conserved DEMs evolved different regulating roles indeed. In the Introduction Page 2, Line 69, we tried to explain the reasons for the study of miR5488.

2) Figure 2 is still a nightmare. IT has only 3 pannels, but the figure legend describes up to (an absent) panel D. IN fact, the figure has been changed from the initial manuscript, but the legend was maintained. It is not described what are the parameters resumed under the title "seed setting rate" in this figure. The relative expression is relative to what? I am still unable to understand what this figure es presenting (I can only guess the intention of authors). Moreover, possible ideas to understand it are on the cover letter but not in the manuscript. Hence, any future reader may have the same concerns than myself, making it unacceptable in the current form.

Response: Thanks for your kind suggestions, we have modified the legend to Figure 2.

3) Concerning point 6 in the cover letter, I was unable to detect the changes without spending an unnecessary time looking for differences among PDFs. In fact, I detected many many unnoticed changes with respect to the previous manuscript regarding text, figures, headers, etc.

Response: Thanks for your kind suggestions. We are very sorry to have your times, since we made great changes in the first manuscript and neglected to mark the all changes. Now, we mark the changes in red or blue, hoping yor can notice easily.

4) Concerning the sentence in lines 144-145 "The percentage of abnormal ovaries amounted to 48% in miR5488-OE-1 and 53% in miR5488-OE-2 lines, respectively", are these values statistically different? I do now think so because the difference in % is very small. Hence, maybe this miRNA is not affecting the ovary development.

Response: Thanks for your kind suggestions. You are right in judging the small difference between 48% in miR5488-OE-1 and 53% in miR5488-OE-2 lines. We evaluated the seed-setting rate of miR5488-OE-1 and miR5488-OE-2 lines, respectively. Thus, we also observed the abnormal ovaries of both transgenic lines, making good explanation for the decreased value of the seed-setting rate. The percentage of abnormal ovaries in miR5488-OE lines were significantly different than that of WT (Page 4, Line 147). So, it was suggested that miR5488 affects the ovary development.

5) Lines 157-160: authors must present numeric evidences that demonstrate that LOC_Os12g41950 is the "authentic target" of their miRNA. Moreover, use only one name for genes, since in the same sentence they indicate that the LOC gene is also called OsARF25, which is the name preferentially used in the manuscript.

Response: Thanks for your good suggestions. In the psRNATarget tool, the target gene is shown by RGAP locus id, such as LOC_Os12g41950, of which gene symbol is OsARF25. Thereforece, LOC_Os12g41950 was firstly listed according to the prediction by psRNATarget tool. Since OsARF25 was presented often, the change was made by typing OsARF25 (LOC_Os12g41950 ) on Page 14, Line 161.

6) In Fig. 5, it is not explained what is the "relative expression". This is absolutely required to know whether the expression of the interesting gene increases or decreases.

Response: Thank you very much for your remider. The phrases of "relative expression" were exhibited in Fig 1, Fig 2C and Fig .4, but not in Fig. 5. The interesting genes or miRNA were clearly indicated.

7) Concerning section 2.5, a miRNA cannot cleave any mRNA. Hence, this section lacks of scientific support. Perhaps authors would like to say that the miRNA guides/marks/selects the mRNA for cleavage. The precise cleavage site described is guided by the miRNA, but not cleaved by the miRNA.

Response: Thank you very much for strong comment and making us understood that the molecular progress, by which miRNAs exert their regulator roles. The reversion was made on Page 6, Line 176.

8) The discussion is a mere review of what are miRNAs doing in their model with only a few words relating to miR5488. This is an important issue about the relevance of this work. 

Response: Your comment is very important. Though miR5488 was screen as DEMs in different papers with RNA-seq, the data about biological function of miR5488 remained unavailable. In this study, we generated transgenic lines of miR5488, and it was lucky to find the alternation of seed-setting rate and embryo sacs structure and proved the down-regulated gene OsARF25. We should have carried out more complicated experiments about miR5488 and OsARF25, so that we could proposed the some type of working model about miR5488

9) English review is still required. I detected many errors even if I am not a native speaker.

Response: Thank you very much for your advice. We tried our best to improve the manuscript and made some changes to the manuscript. These changes will not influence the content and framework of the paper. And the changeswere marked in red and blue in the revised manuscript. We appreciate for Reviewers’ warm work and hope that the correction will meet with approval.

Reviewer 2 Report (New Reviewer)

Comments and Suggestions for Authors

The manuscript describes the function of miRNA 5488 in rice embryogenesis. Phenotypes of overexpression lines indicate that miRNA5488 is involved in regulating embryogenesis. In addition, expression analysis and reporter assays indicate that the target of miRNA5488 is OsARF25. In summary, this manuscript provides novel insights into the function of miRNA5488 in rice embryogenesis. However, it has not been pursued whether the phenotype in miRNA5488 overexpressing lines is due to silencing of OsARF25. Therefore, the function of OsARF25 is unclear, and additional experiments should be conducted to determine whether OsARF25 is involved in embryogenesis and primary root elongation. The issues listed below should be addressed before publication.

1)     It is unclear whether OsARF25 is involved in the phenotype of miRNA5488 overexpressing lines. It would be desirable to pursue whether OsARF25-repressed or mutant strains show a phenotype similar to that of miRNA5488 overexpressing lines. Alternatively, it is possible to examine whether overexpression of OsARF25 in the miRNA5488 overexpressing line restores the phenotype.

2)     Why can the authors conclude that roots of miRNA5488 overexpressing plants are less sensitive to IAA? (line 227-228)

When 10 µM IAA is added, the overexpression lines appear to be less sensitive than the wild-type lines, as the suppression of elongation of the primary roots is reduced. However, when 0.1 or 1 µM IAA is added, the overexpression lines do not show the enhanced primary root elongation in the wild-type lines. Instead, primary root elongation is suppressed in the overexpression lines. Is there a possible theoretical explanation for this? The manuscript needs to observe the dose dependency more closely or analyze the expression of auxin-responsive genes in roots to discuss the sensitivity of IAA.                                            

Author Response

Dear Reviewers,

Thanks very much for taking your time to review this manuscript. I really appreciate all your comments and suggestions! Please find my itemized responses in below and my revisions/corrections in the re-submitted files.

Reviewer 2

1) It is unclear whether OsARF25 is involved in the phenotype of miRNA5488 overexpressing lines. It would be desirable to pursue whether OsARF25-repressed or mutant strains show a phenotype similar to that of miRNA5488 overexpressing lines. Alternatively, it is possible to examine whether overexpression of OsARF25 in the miRNA5488 overexpressing line restores the phenotype.

Response: Thank you very much for your comment, which provides a cue about further experimental design to elucidate the relationship between miR5488 and OsARF25 in detail. In this study, it was novel to find the alternation of seed-setting rate and embryo sacs structure in miR5488-OE lines, and to reveal the down-regulation of target OsARF25 by miRNA5488. Based on these results, the over-expression of OsARF25 lines will be generated and then to used to crossed with miR5488-OE lines in future studies. The knock-down or knock-out of  OsARF25 lines will be generated, too.

2)     Why can the authors conclude that roots of miRNA5488 overexpressing plants are less sensitive to IAA? (line 227-228)

When 10 µM IAA is added, the overexpression lines appear to be less sensitive than the wild-type lines, as the suppression of elongation of the primary roots is reduced. However, when 0.1 or 1 µM IAA is added, the overexpression lines do not show the enhanced primary root elongation in the wild-type lines. Instead, primary root elongation is suppressed in the overexpression lines. Is there a possible theoretical explanation for this? The manuscript needs to observe the dose dependency more closely or analyze the expression of auxin-responsive genes in roots to discuss the sensitivity of IAA.    

Response: We were firstly astonished with the distinct change of the primary root length were observed in WT and miR5488-OE lines, after being subject to different concentration of IAA. In WT line, the length of primary root were increased significantly at low level 0.1 µM of IAA, and were not changed significantly at at middle level 1 µM of IAA, but were decreased largely at high concentration of 10 µM IAA. It was implied that there might be a negative feedback mechanism about the effect of IAA on root length. The negative feedback mechanism is common in phytohormone-related regulation. In miR5488-OE, the length of primary root were decreased significantly at three levels of IAA, it might imply that the increased expression level of miR5488 might destroy such a negative feedback mechanism to repress the growth of primary root slightly. We tested the response of miR5488-OE to IAA in order to indicate miR5488 might affect the development of embryo sac by auxin signal, since the down-regulation of OsARF25 by miR5488 was revealed here. Your suggestion is very important, the relevant experiments will be carried out in future study. 

Round 2

Reviewer 1 Report (Previous Reviewer 2)

Comments and Suggestions for Authors

The new version of the manuscript answered all my concerns.

Reviewer 2 Report (New Reviewer)

Comments and Suggestions for Authors

The authors have addressed the concerns raised in previous reviews.

This manuscript is a resubmission of an earlier submission. The following is a list of the peer review reports and author responses from that submission.

Round 1

Reviewer 1 Report

Comments and Suggestions for Authors

In this work, Guo et al. have attempted to characterize the function of a miRNA in rice under various growth conditions. To achieve that, the authors generated rice lines overexpressing the target miRNA and that where it is silenced, and compared various parameters with the wild type. Although this study generates information that could be of interest to the readers, this manuscript still requires substantial changes and additional data to comprehend and bring more evidence that can support the claim by the authors.

I have raised concerns in the manuscript that are highlighted in the PDF version of the manuscript attached here. Considering the complexity of the study, the authors must add a conclusion section where they portray the significance of their findings and give a take-home message for the readers.

Comments on the Quality of English Language

I found some grammatical errors in the manuscript that require close attention. The entire manuscript should be revised to English.

Reviewer 2 Report

Comments and Suggestions for Authors

The manuscript of Guo et al focuses on the study of microRNA based on its over-expression and suppression using basic molecular methods. However the manuscript is very difficult to follow since mentions to figures is completely messed up, the English contains many error (even detected by a non-native English speaker like me!), results contain methods and introduction, as well as references, etc. Hence, even if the work would be interesting, I have to reject it in the current form. Here I give authors some (there are many more!) concerns to illustrate my decision.

- The introduction does not explain why authors choose MIR5488 for their study.

- Why most microRNA are named 'miRNAxxx' while the one studied in this work is many times (but not always) called as MIRNA5488'? Authors must be consistent.

- Please, give more details about the rationale of suggestion of page 2 deduced from Figure 2: It was suggested that OsmiR5488 might affect the fertility of mature pollen and embryo sacs to regulate the seed setting rate. I think that this suggestion goes beyond the results.

- Authors must explain how was calculated the relative expression of Fig 2, since it is crucial to know if OE and STTM strains are what they claim.

- Mention to figures is a complete mess!!! I was unable to link text with figures. For example, the first figure mentioned after figure 2 is Fig 5 while Figs 3 and 4 were not mentioned before. But the cited Figure 5 has many panels and I think it is really Fig 3. However, the panels mentioned in the text do not correspond to  those in this figure 3. In section 2.5, mentions to Fig 6 seem to be to Fig 5. And so on...

- Section 2.1 stars with some idea that must be in the Introduction.

- Section 2.4 starts explaining the method, instead of place methods in the correct section.

- It was impossible to me to evaluate if the experimental approach is right and if conclusions obtained from experiments and figures can explain their proposals. I cannot follow the Discussion without a clear idea of results. 

- After rethinking and rewriting the manuscript, maybe authors must also rethink if IJMS is the appropriate journal for their results.

Comments on the Quality of English Language

Only a few examples:

- regulat or → regulator

- specie-specific → species-specific 

- a auxin → an auxin, but two words later it is mentioned «genes». Please be consistent: one gene or many genes?

- 3’ -amplifification: remove the extra space

- There are many double or triple white spaces in many instances of the document.

- dicot → dicotyledones or Dicotyledoneae. The same for monocots.

- Arabidopsis → in italics, please.

- 'Ubiquitinly' does not exist in English, use 'ubiquitously'

- throug ???
